# The Study on the Lasing Modes Modulated by the Dislocation Distribution in the GaN-Based Microrod Cavities

**DOI:** 10.3390/nano13152228

**Published:** 2023-08-01

**Authors:** Yuyin Li, Peng Chen, Xianfei Zhang, Ziwen Yan, Tong Xu, Zili Xie, Xiangqian Xiu, Dunjun Chen, Hong Zhao, Yi Shi, Rong Zhang, Youdou Zheng

**Affiliations:** Key Laboratory of Advanced Photonic and Electronic Materials, School of Electronic Science and Engineering, Nanjing University, Nanjing 210093, China; dg20230024@smail.nju.edu.cn (Y.L.); 602022230055@smail.nju.edu.cn (X.Z.); dg21230063@smail.nju.edu.cn (Z.Y.); mg21230048@smail.nju.edu.cn (T.X.); xzl@nju.edu.cn (Z.X.); xqxiu@nju.edu.cn (X.X.); djchen@nju.edu.cn (D.C.); zhaohong@nju.edu.cn (H.Z.); yshi@nju.edu.cn (Y.S.); rzhang@nju.edu.cn (R.Z.); ydzheng@nju.edu.cn (Y.Z.)

**Keywords:** GaN, microrod cavity, low-threshold, dislocation, whispering gallery mode

## Abstract

Low-threshold lasing under pulsed optical pumping is demonstrated in GaN-based microrod cavities at room temperature, which are fabricated on the patterned sapphire substrates (PSS). Because the distribution of threading dislocations (TDs) is different at different locations, a confocal micro-photoluminescence spectroscopy (μ-PL) was performed to analyze the lasing properties of the different diameter microrods at the top of the triangle islands and between the triangle islands of the PSS substrates, respectively. The μ-PL results show that the 2 μm-diameter microrod cavity has a minimum threshold of about 0.3 kW/cm^2^. Whispering gallery modes (WGMs) in the microrod cavities are investigated by finite-difference time-domain simulation. Combined with the dislocation distribution in the GaN on the PSS substrates, it is found that the distribution of the strongest lasing WGMs always moves to the region with fewer TDs. This work reveals the connection between the lasing modes and the dislocation distribution, and can contribute to the development of low-threshold and high-efficiency GaN-based micro-lasers.

## 1. Introduction

In the heteroepitaxy growth of GaN-based materials, it has been well known that threading dislocations (TDs) are inevitable [1]. Compared to GaAs family semiconductors, GaN family semiconductors are relatively less sensitive to TDs, but the high TD density still limits the performance of GaN devices in the field of relatively strong photon–electron coupling, such as lasing. Thus, the analysis of GaN lasing performance dependent on TDs has important research significance for improving the overall performance of GaN lasing devices.

In recent years, patterned sapphire substrates (PSS) have been widely used for high-quality GaN growth, especially for the growth of GaN-based light emitting diodes (LEDs). The patterned sapphire substrate can effectively reduce the dislocation density generated during the growth of GaN, so as to improve the radiative recombination quantum efficiency and light extraction efficiency of LEDs [2,3,4]. Although the effect of TDs on GaN-based LEDs is slightly smaller, the “bad” effect of TDs has been studied by many researchers in electronic devices [5,6]. Therefore, for advanced lighting devices, such as lasers, the dislocation cannot be treated as it is in LEDs and may be a key factor for lasing. In one typical laser device, due to the large active region, the TDs will give a total effect; for the micro/nano cavity, the small active region gives the opportunity to investigate the TD effect by distinguishing different TD distributions in the micro/nano cavities. However, the influence of dislocation on micro/nano-laser performance has not been studied deeply.

A laser requires a gain medium and a resonant cavity to produce coherent photons through stimulated emission [7]. WGM microcavities have been demonstrated as excellent candidates for constructing low-threshold and narrow-linewidth lasers [8,9]. For example, Mi et al. demonstrated a low-threshold planar laser based on high-quality single-crystalline hexagonal CdS nanoplatelets using a self-limited epitaxial growth method [10]. Sellés et al. showed a deep-UV nitride-on-silicon microdisk laser with a series of WGMs circulating at the periphery of the disk [11]. The circular microcavities are prepared by the top-down method; their performance has been limited by variabilities in the material growth and processing techniques [12]. Due to the existence of lasing mode competition, single mode or multi-mode lasers have been achieved [13,14,15]. The optical field of WGMs of the circular microcavity is uniformly distributed along the circular sidewall. It is an interesting question whether the density of the dislocation in the optical field distribution region has any effect on the performance of the lasing behavior.

In this paper, optically-pumped lasing from InGaN/GaN MQW microrod cavities is demonstrated based on a standard blue GaN–LED wafer. We fabricated microrod cavities with diameters of 2, 3, and 4 μm at different positions on the TPSS by the top-down method so that we could distinguish the distribution of the dislocation in the microrod cavities. Lasing emission was observed under pulsed optical pumping. The FDTD method was used to calculate the electric field distribution in all microrod cavities.

## 2. Materials and Methods

The microrod cavities studied in this work were fabricated on a standard blue LED structure grown on a (0001) PSS sapphire substrate using metal–organic chemical vapor deposition (MOCVD). A 3.5 μm-thick un-doped GaN (u-GaN) layer was grown first on the substrate. A 2 μm n-GaN layer with a Si-doping concentration greater than 10^19^/cm^3^ was grown above the u-GaN layer and followed by 10-period InGaN/GaN (3 nm/12 nm) multiple quantum wells (MQWs) with a composition of 13%. Finally, a 50 nm-thick Mg-doped p-GaN with a doping concentration greater than 10^19^/cm^3^ was grown. The fabricating process is illustrated in Figure 1, and the small triangle islands structure on the PSS surface is illustrated in Figure 1e. It is obvious that different locations around the small triangle islands have different dislocation distributions. Then, we selected two positions, one located at the top of each triangle island called position A, and the other located between the triangle islands called position B. Correspondingly, we fabricated microrod cavities at the two positions to explore the influence of dislocation on the lasing behavior.

The microrod cavities were made in the following way. A 600 nm SiO_2_ film was deposited by Plasma Enhanced Chemical Vapor Deposition (PECVD, Oxford, UK) (Figure 1b). The Ni mask was patterned using UV photolithography (Figure 1c). Inductively coupled plasma (ICP, Oxford, UK) was used to etch a SiO_2_ hard mask (Figure 1d). Then, the GaN-based microrod was formed by ICP dry etching (Figure 1e). After that, the remaining SiO_2_ was removed by diluted HF solution (Figure 1f). Finally, surface damages caused by ICP etching were removed by wet etching with a 2 mol/L KOH solution heated to 80 °C for about 40 min. We fabricated the microrod cavities with diameters of 2 μm, 3 μm and 4 μm. Since one group of microrod cavities is located on the top of each triangle island, i.e., position A, we called the microrod cavities 2A, 3A, and 4A. Another group of microrod cavities is located between the triangle islands, i.e., position B, so we called the microrod cavities 2B, 3B and 4B. The SEM images of all samples are shown in Figure 2. Figure 2g shows the topography of the PSS surface; the distance between the centers of the two islands is about 3.8 μm. The circles mark the projection positions of the six samples on the substrate, respectively.

According to previous work, we knew that ICP etching can cause sidewall damage [16,17,18], and the photoresist as a mask will cause the sidewall to tilt [12,19,20]. In this work, Ni hard mask was used instead of a photoresist, leading to smooth and vertical microrods, as shown in Figure 2. These structural features are essential to the formation of the WGMs investigated later.

To pump the microrods and collect emissions, we used a standard confocal Renishaw inVia Reflex μ-PL setup with a 375 nm pulsed laser (115 fs pulse width; 76 MHz repetition rate) and a 50× microscope objective. The detector consisted of a spectrometer and a Peltier-cooled charge-coupled device (CCD). All PL measurements were performed at room temperature.

Finite-difference time-domain (FDTD) simulations were used to identify the visible mode of microcavities with different diameters. According to the material properties, GaN and InGaN materials have TE mode optical gain, which causes the electric fields of dipoles acting as a pumping source to polarize along the road plane.

## 3. Results

The emission spectra with different pumping power densities for 2A/2B/3A/3B and 4A/4B are shown in Figure 3. For all samples, the spectra and the light-out intensity versus pump power curves show lasing behavior. The sharp stimulated emission peaks emerged from the broad spontaneous emission background, which does not shift with the increase of excitation power density. For the samples in positions A and B, it can be seen that the larger the diameter, the larger the difference in the lasing peak wavelength between positions A and B. When the diameter of the microrod increases to 4 μm, the PL spectra of 4A and 4B show a significant difference. The wavelength of the main lasing peak of 4A is 425.98nm, and that of the main lasing peak of 4B is 438.51 nm. Compared with microrods with diameters of 2 μm and 3 μm, the number of lasing peaks decreases, and the main peaks show absolute superiority, as shown in Figure 3i,k.

The thresholds of 2A and 2B are 0.300 kW/cm^2^ and 0.360 kW/cm^2^, respectively. The thresholds of 3A and 3B are 0.343 kW/cm^2^ and 0.405 kW/cm^2^, respectively. We can observe that the samples at position A always show a lower threshold of about 20% than those at position B. The sample of 4A also shows a lower threshold (0.830 kW/cm^2^) than that of 4B (1.071 kW/cm^2^).

The quality factor Q can be calculated from the relation Q=λ/Δλ, where *λ* and Δλ are the central wavelength and the full width at half maximum (FWHM) of the lasing peak, respectively. The Q factors for the main lasing peak of the 2A and 2B are about 1208 (λ=435.08 nm) and 1502 (λ=432.65 nm), respectively. The Q factors of 3A and 3B were 1313 (λ=428.24 nm) and 1470 (λ=429.31 nm), respectively. The Q factors of the main peak of 4A and 4B were 1521 and 1414, respectively. The laser parameters of all samples are listed in Table 1.

All these lasing modes have been identified as WGM lasing in the later simulation results. This is because of the smooth and vertical sidewall of these microrods, which allows optical confinement in the MQW plain. Because our microrods diameter is small, if the sidewall is tilted, light will not be well confined in the MQW plain, and the overlap between the axial electric field and the active region will be reduced, which will reduce the efficiency of photon–electron coupling and reduce the probability of laser generation. At the same time, the light propagation will randomly go to the bottom of the microrods, and the resonant wavelengths will be random as well [16]. This makes it difficult to get a standard WGM laser. In Figure 3, the lasing peaks do not shift under the varying excitation power density and always exist over the test range, which is consistent with the WGMs obtained by later FDTD simulations.

Figure 4 summarizes the threshold and slope efficiency changes of the microrods with varying diameters. The slope efficiency is obtained from the slope of the curve of light intensity with the pump power in Figure 3. It can be seen that for positions A or B, the threshold increases with the increase of the microrod diameter. This is because the larger the diameter, the longer the propagation path of light in the microrods, the larger the optical scattering and loss, resulting in a larger threshold [16]. For the samples with the same diameters, those at position A show a lower threshold and higher efficiency than those at position B, but the microrods with a diameter of 4 μm have a bigger difference. The larger diameter is the most possible reason, because 4 μm is larger than the distance between the islands (3.8 μm), which means that 4B can cover the top of three islands and suffer more influence from the dislocations. We will explain this with FDTD simulations in detail below.

The use of PSS substrate technology to reduce the dislocation density generated during GaN epitaxial growth has been extensively studied, and the growth process and dislocation behavior of GaN grown on PSS have also been intensively studied; the dislocation evolution process of GaN grown on PSS substrate shows consistency [21,22,23,24,25,26]. According to the growth mechanism of a GaN epitaxial wafer on PSS [4,21,22,23,24,25,26], the GaN will start its growth from the flat surface (FS) areas between the islands and fill up the space between the islands. The growth in the FS areas will cause the dislocations to extend to the epitaxial layer surface almost vertically, because GaN growth rate on the bottom of the FS is much faster than on the top of the pattern. Finally, the GaN layer gradually overgrows the islands. However, the covering growth on the islands is a lateral overgrowth; thus, the dislocations on the tilted surface extending direction will bend 90°, and coalesce near the summit the islands’ top [21,22]. During the dislocation coalescence, when some dislocations meet from both sides, they can be annihilated in the top region of the island. Therefore, most dislocations are from the FS areas and there are some mixed dislocations extending from the top of the islands [4,21,25]. Therefore, the dislocations have regular distribution in the GaN epitaxial layer, as shown in Figure 5d. In the FS region, the dislocations extend directly to the GaN surface; on the tilted surface, they bend due to lateral overgrowth and coalesce near the summit, finally extending to the surface [22,23,24].

The WGM lasing modes of the microrod cavities were calculated by the 2D-FDTD method. The radial electrical profile at 435.08 nm of 2A is shown in Figure 5a. This lasing mode is the first-order WGM in terms of radial mode number. The lasing mode of 3A is the third-order WGM at 428.24 nm, shown in Figure 5b. However, the lasing mode of 4A is the eighth-order WGM at 425.98 nm, shown in Figure 5c. The vertically shaded areas in Figure 5d represent the corresponding microrods. It is clear that the TDs only exist in the center area of 2A, and a few TDs are captured by the sidewall of 3A, so the field patterns of their lasing modes are mainly distributed near the edge of the microrods, where there are few TDs. For 4A, because its size is larger than one island, it covers the FS areas, so a large number of TDs have been involved in the microrod’s sidewall. However, the TDs around the sidewall region will cause damage to the electron radiative recombination, reducing the electron–photon coupling efficiency. Therefore, the light path will pass through more no-TD areas, that is, push the lasing region into the inner area. This leads to the occurrence of a higher-order lasing mode in 4A, where the location of the optical field is almost free of TDs.

When we look at the samples in position B from the 2D-FDTD calculations, the lasing mode of 2B is also the first-order WGM at 432.65 nm, as shown in Figure 6a. The lasing mode of 3A is also the third-order WGM at 429.31 nm, shown in Figure 6b. It can be seen from Figure 6d that the TD distribution in 2B and 3B is similar to that in 2A and 3A. In position B, both samples have a large number of TDs in the center area. Thus, the field patterns of the strongest lasing modes are the same. However, the main lasing mode of 4B is fourth-order WGM; as shown in Figure 6c, the field patterns of the strongest lasing mode are closer to the sidewall than those in Figure 5c. We noticed that the dislocation distribution of 4B was significantly different from that of 4A. Due to the nature of the dislocation distribution in the epitaxial layer, fewer TDs can be included in the sidewall of 4B than in that of 4A. At the same time, there are more dislocations in the center of the microrod, which allows the light to circulate near the sidewall, meaning that the TDs push the laser region not too far away from the sidewall. Therefore, it is clear that the occurrence of actual lasing modes depends on the dislocation distribution in the structure.

Here we can see that 2B and 3B both contain a large number of TDs in the center. Under the same pumping conditions, the total number of photons generated by radiation recombination in the entire MQW plain will be less than in 2A and 3A, which will reduce the number of eligible photons that are fully reflected along the boundary, so the thresholds of 2B and 3B will be higher than of 2A and 3A, respectively. The center of 4B also contains a large number of TDs. In addition, the sidewall occupies the top area of several islands at the same time, which also contains more dislocation. The collective effect of these two factors leads to a lower total number of photons in the MQW plain of 4B than that of 4A, which also leads to a lower number of photons traveling along the edge of 4B, which leads to a higher threshold of 4B.

The field patterns of the strongest lasing mode always coincide with the lower TD density region, and the distribution of dislocations in the microrods will affect the dominant lasing mode. It can be concluded that the lower the TD density in the WGM mode region, the lower the threshold and the higher the efficiency.

## 4. Conclusions

In summary, the lasing behavior and mode distribution in the GaN-based microrod cavities with different dislocation distributions were studied in this paper. The threshold and slope efficiency of the microrod laser are closely related to the dislocation distribution. Lower dislocation density is beneficial for achieving higher performance lasing. This work shows that the field patterns of the strongest lasing mode always coincide with the lower TD density region, and the distribution of dislocations in the microrods will affect the dominant lasing mode. The 2 μm-diameter microrod cavity has a minimum threshold of about 0.3 kW/cm^2^. As a result, one reasonable control of dislocation distribution can effectively reduce the impact of dislocation on lasing performance and control the final lasing mode. This result can contribute to the development of low-threshold, high-efficiency, and mode-controlled GaN-based WGM micro-lasers.

## Figures and Tables

**Figure 1 nanomaterials-13-02228-f001:**
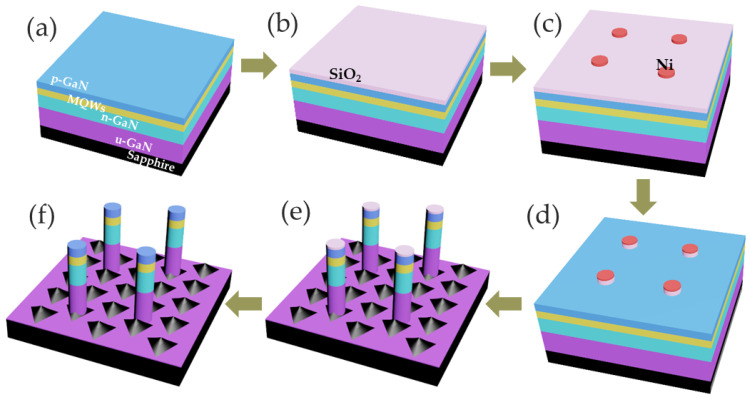
Fabrication process of the microrod cavities. The original epitaxial wafer (**a**), the SiO_2_ deposition (**b**), the Ni hard mask formed by photolithography and lift-off (**c**), etching of SiO_2_ (**d**), inductively coupled plasma itching of the wafer (**e**), and removal of the SiO_2_ (**f**).

**Figure 2 nanomaterials-13-02228-f002:**
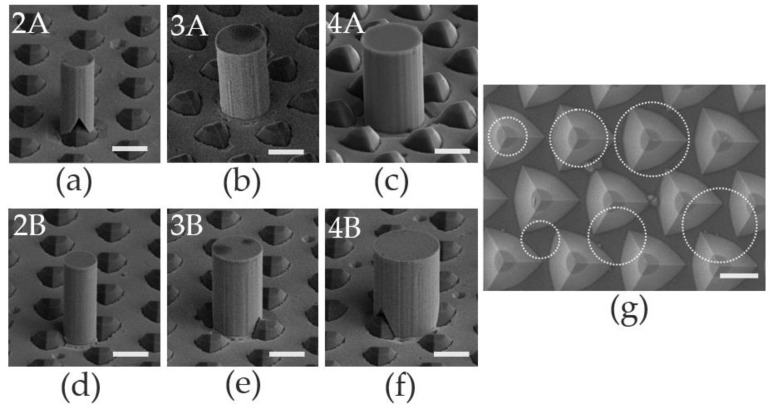
SEM images of microrod on the top of the triangle island with diameter of 2 μm, 2A (**a**), 3 μm, 3A (**b**), and 4 μm, 4A (**c**); between the triangle islands with diameter of 2 μm, 2B (**d**), 3 μm, 3B (**e**), and 4 μm, 4B (**f**). The morphology of the TPSS (**g**), in which the circles mark the projection positions of our six samples on the substrate. The scale bars represent 2 μm.

**Figure 3 nanomaterials-13-02228-f003:**
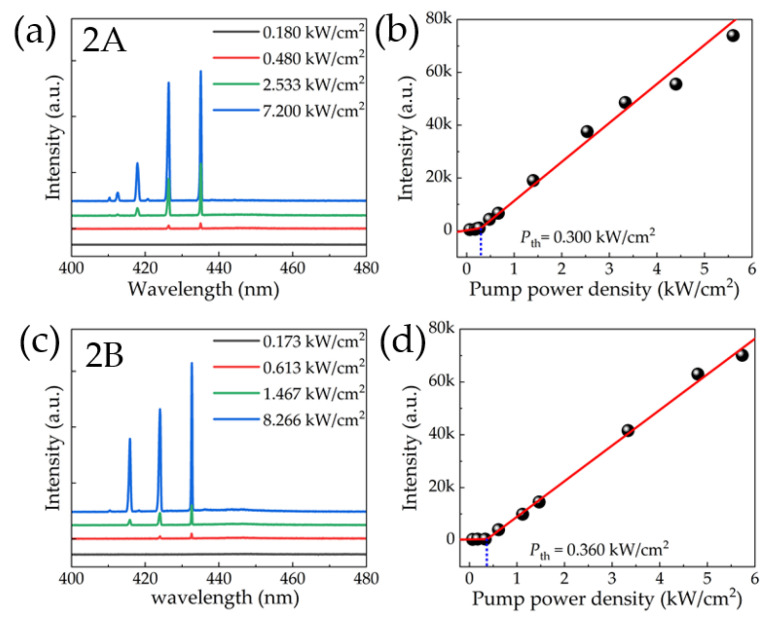
PL from (**a**) 2A, (**c**) 2B, (**e**) 3A, (**g**) 3B, (**i**) 4A, and (**k**) 4B. The relationship between PL intensity and pump power density are plotted in (**b**,**d**,**f**,**h**,**j**,**l**) for the main peak, respectively.

**Figure 4 nanomaterials-13-02228-f004:**
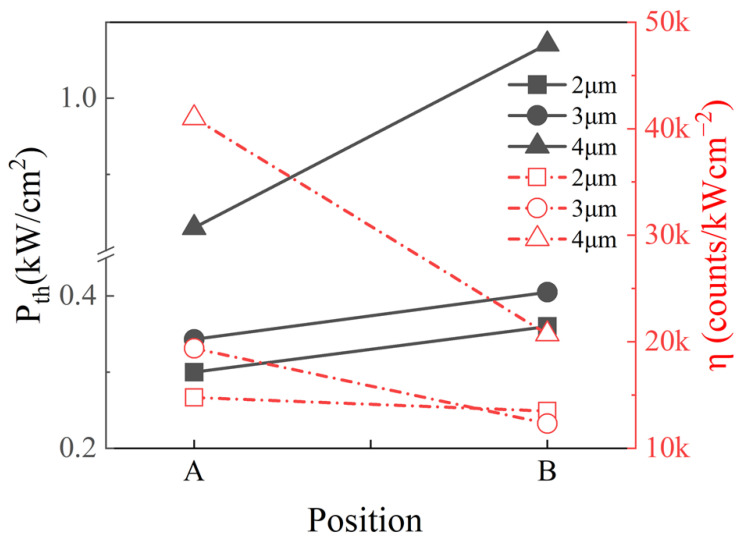
The threshold and slope efficiency of the microrod cavities with different diameter at position A and B.

**Figure 5 nanomaterials-13-02228-f005:**
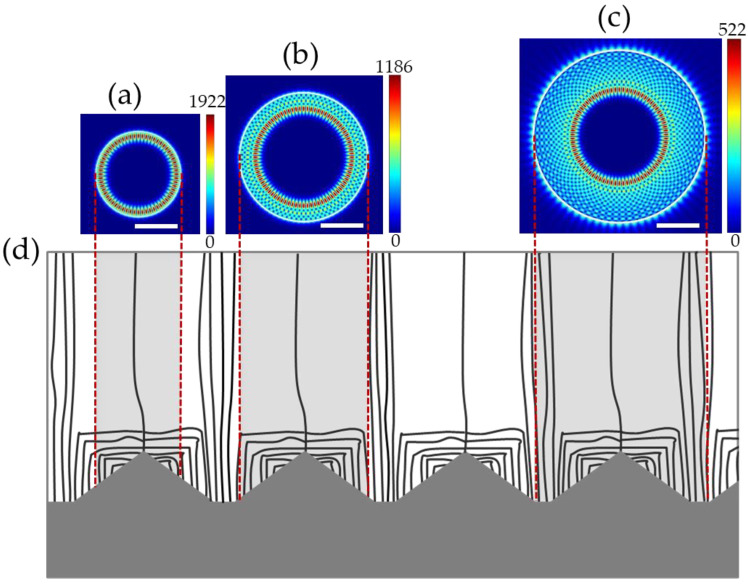
At position A, FDTD-simulated electric field intensity profiles (|*E*|) of 2 μm-diameter microrod at 432 nm (**a**), 3 μm-diameter microrod at 428 nm (**b**), and 4 μm-diameter microrod at 425 nm (**c**); the scale bars represent 1 μm. Schematic diagram of dislocation distribution in the GaN grown on the PSS substrate (**d**).

**Figure 6 nanomaterials-13-02228-f006:**
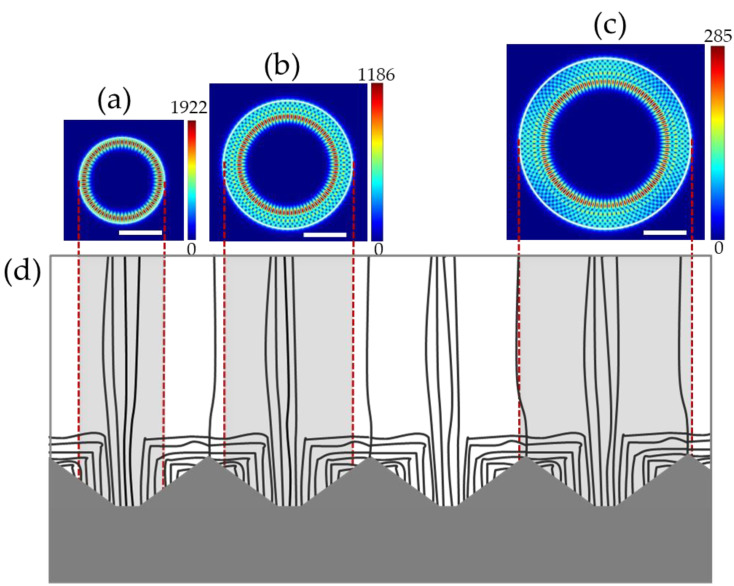
At position B, FDTD-simulated electric field intensity profiles (|*E*|) of 2 μm-diameter microrod at 432 nm (**a**), 3 μm-diameter microrod at 428 nm (**b**), and 4 μm-diameter microrod at 438 nm (**c**); the scale bars represent 1 μm. Schematic diagram of dislocation distribution in the GaN grown on the PSS substrate (**d**).

**Table 1 nanomaterials-13-02228-t001:** Dominate laser parameters of all samples.

	2A	2B	3A	3B	4A	4B
P_th_ (KW/cm^2^)	0.300	0.360	0.343	0.405	0.830	1.071
Wavelength (nm)	435.08	432.65	428.24	429.31	425.98	438.51
FWHM (nm)	0.36	0.29	0.33	0.29	0.28	0.31
Q	1208	1502	1313	1470	1521	1414

## Data Availability

All data, theory detail, and simulation detail that support the findings of this study are available from the corresponding authors upon reasonable request.

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
