# Peer review of "The Study on the Lasing Modes Modulated by the Dislocation Distribution in the GaN-Based Microrod Cavities"

_nanomaterials, 2023, doi:10.3390/nano13152228_

Round 1
Reviewer 1 Report
- The electric field intensity distribution for the 2A and 2B are identically not overlapping with the vertically extending TD, however, we observe still a difference in the Pth.
- The electric field intensity distribution for the 3A is even more overlapping with the vertically extending TD, than for the 3B, however, the difference in the Pth is in favor of the 3A.
- The photonics of how the TDs are “pushing lasing region into inner area” or “to push the laser region not too far away from the sidewall” should be explained. If the pumping conditions are the same, higher losses at the sidewalls for the 4A should lead to higher Pth than for 4B, which is not the case.
The grammar should be checked.
Author Response
Point 1: The electric field intensity distribution for the 2A and 2B are identically not overlapping with the vertically extending TD, however, we observe still a difference in the Pth.
Response 1: We appreciate the reviewer's comment. Under the condition of optical pumping, due to the large number of dislocations in the center of the 2B, the total number of photons generated by the radiation recombination of the entire MQW plain is less than that of 2A, so the photon density that can be periodically reflected along the sidewall will lower than 2A, resulting in 2B exhibiting a lower threshold.
Point 2: The electric field intensity distribution for the 3A is even more overlapping with the vertically extending TD, than for the 3B, however, the difference in the Pth is in favor of the 3A.
Response 2: We thank the reviewer's input. The sidewall of 3A has some dislocations, while the center of 3B contains many dislocations, resulting in a larger dislocation density in the entire MQW plain of 3B than 3A, and lastly, 3B exhibits a higher threshold.
Point 3: The electric field intensity distribution for the 3A is even more overlapping with the vertically extending TD, than for the 3B, however, the difference in the Pth is in favor of the 3A.
Response 3: We thank the reviewer for the suggestion.
Under the condition of optical pumping, a large number of photons are generated due to the radiation recombination in MQW plain. When the sidewall of the microrod contains a large number of dislocations will cause damage to the electron radiative recombination, the photons density at the sidewall is lower than inner area, so the photon travel in a larger range in the interior. When the dislocation density in the center of the microrod is higher than that at the sidewall, the photon density near the sidewall is higher, so the photons are easier to circulate within the range not far from the sidewall. We have also added explanations in the revised manuscript.
The 4B's center contains a large number of dislocations, and the sidewalls occupy multiple islands' top where also contains much mixed dislocations. This results in the radiation recombination in the entire MQW plain producing fewer photons than 4A, and ultimately the photon numbers propagating along the microcavity boundary will lower, so 4B has a higher threshold than 4A.
Other problems such as the language and grammar mistakes have been rectified in the revised manuscript.

Reviewer 2 Report
In this manuscript, “The study on the lasing modes modulated by the dislocation distribution in the GaN-based microrod cavities,” the authors present the connection between lasing modes and the corresponding dislocation distribution. Overall, this manuscript has a strong potential for another review round after applying the issues and addressing the shortcomings listed below:
1-The authors should polish/revise some grammatical mistakes and typos along the manuscript. I invite the authors to read their manuscript carefully and make the required changes where necessary.
2-Please increase the size of the text provided in the figures (where necessary).
3-Please increase the thickness of the lines within the figures (where necessary).
4-In the Introduction section, while discussing recent developments in the field of lasing modes, the following works should also be considered and cited to give a more general view to the possible readers of the work: [(i) Electrically Driven Hot-Carrier Generation and Above-Threshold Light Emission in Plasmonic Tunnel Junctions, Nano Letters 20, 6067-6075 (2020); (ii) Room-temperature exciton-polariton and photonic lasing in GaN/InGaN core-shell microrods, Physica Scripta 98, 074001 (2023)].
5-In Figures 1b and 1c, it is hard to read “SiO2” and “Ni” due to white color. Please change the white color.
6-In Figures 5a-5c, please put the corresponding color bars. Also, please put more detailed discussion about Figure 5d in the manuscript.
7-In Figures 6a-6c, please put the corresponding color bars. Also, please put more detailed discussion about Figure 6d in the manuscript.
N/A.
Author Response
Point 1: The authors should polish/revise some grammatical mistakes and typos along the manuscript. I invite the authors to read their manuscript carefully and make the required changes where necessary.
Response 1: Agreed. We have inspected the manuscript carefully and rectified the grammatical mistakes and typos in the revised manuscript.
Point 2: Please increase the size of the text provided in the figures (where necessary).
Response 2: Agreed. Some of the size of the text in Figures 2, 3, 5, and 6 have been increased and can be seen more clearly in the revised manuscript.
Point 3: Please increase the thickness of the lines within the figures (where necessary).
Response 3: Agreed. The thickness of the lines in Figure 5, 6 have been modified, and also updated in the revised manuscript.
Point 4: In the Introduction section, while discussing recent developments in the field of lasing modes, the following works should also be considered and cited to give a more general view to the possible readers of the work: [(i) Electrically Driven Hot-Carrier Generation and Above-Threshold Light Emission in Plasmonic Tunnel Junctions, Nano Letters 20, 6067-6075 (2020); (ii) Room-temperature exciton-polariton and photonic lasing in GaN/InGaN core-shell microrods, Physica Scripta 98, 074001 (2023)].
Response 4: We thank the reviewer for the suggestion. We read these two papers carefully, paper (i) is less relevant to our content, and paper (ii) is cited in the revised manuscript [Ref.15].
Point 5: In Figures 1b and 1c, it is hard to read “SiO2” and “Ni” due to white color. Please change the white color.
Response 5: Agreed. In Figure 1b and 1c, “SiO2” and “Ni” have been modified. Please see Figure1 in the revised manuscript.
Point 6: In Figures 5a-5c, please put the corresponding color bars. Also, please put more detailed discussion about Figure 5d in the manuscript.
Response 6: We thank the reviewer for the suggestion. The color bars have been added in Figures 5a-5c. In the revised manuscript, the evolution process of the dislocation of GaN grown on PSS substrate is described in more detail, and the distribution of the TDs in Figure 5d is also explained.
As follows:
According to the growth mechanism of a GaN epitaxial wafer on PSS [4, 21-26], the GaN will start its growth from the flat surface (FS) areas between the islands and fill up the space between the islands. The growth in the FS areas will cause the dislocations extending to the epitaxial layer surface almost vertically. Because GaN growth rate on the bottom of FS is much faster than on the top of the pattern. Finally, the GaN layer gradually overgrew the islands. However, the covering growth on the islands is a lateral overgrowth; thus, the dislocations on the tilted surface extending direction will bent 90°, and coalesced near the summit the islands’ top [21,22]. During the dislocation coalescence, when some dislocations meet from both sides, they can be annihilated in the top region of the island. So, most dislocations are from the FS areas and there are some mixed dislocations extending from the islands’ top [4, 21, 25]. Therefore, the dislocations have regular distribution in the GaN epitaxial layer, as shown in Figure 5d. In the FS region the dislocations extend directly to the GaN surface, on the tilted surface they bend due to lateral overgrowth and coalesced near the summit, finally extending to the surface [22-24].
Point 7: We thank the reviewer for the suggestion. The color bars have been added in Figures 6a-6c. Figure 6d and Figure 5d have the same description as illustrated in the revised manuscript.
Response 7: We thank the reviewer for the suggestion. The color bars have been added in Figures 6a-6c. Figure 6d and Figure 5d have the same description as illustrated in the revised manuscript.

Reviewer 3 Report
In this work, the authors fabricated InGaN-GaN MQW microrod cavities to study how lasing modes correspond with threading dislocation density distribution. The draft is generally well written and the data presentation is clear. There are some critical issues the authors need to address before the paper can be recommended for publication.
1. (line 76-79) More details of the epitaxial growth process are needed. What are the doping levels of the n-type and p-type GaN? What is the structure of the MQW?
2. (line 139) Please avoid using phrases like “it seems that…”
3. (line 138-147) I recommend the authors present these numeric values in the form of a table, listing the threshold, peak wavelength, FWHM, and the Q factor for each sample.
4. (line 148) Typo? “later” -> “laser”
5. (line 151) Could the authors provide a brief explanation and appropriate references for this statement? The readers may not be familiar with the effect of having tilted sidewalls.
6. (line 152) What does “regular and stable” mean? Please avoid using descriptive words. The statement needs to be scientifically precise and add values to the readers understanding of the subject matter.
7. (line 160) Please delete the word “obviously”
8. (line 170-224) The logic of this section needs clarification. There are 3 things involved in this discussion:
A. Lasing modes of the MQW structures the authors fabricated.
B. Lasing modes calculated by FDTD.
C. TD distribution (Figure 5d).
We can only consider A as facts among these three, as it is experimentally measured. From my understanding of how the current draft is written, the authors made the assumption that C is true, then incorporated C in FDTD simulation and obtained results B. However, unless C is verified to be true, the comparison between A and B would be invalid. In other words, I understand that it makes sense that stronger lasing modes would correlate with regions with lower TD density, but we cannot say that TD density is “the” root cause. There could well be other factors that would affect lasing modes.
My suggestion: One way of making this discussion logically sound is by proving C a fact — For example, it can be proved by cross-sectional TEM imaging of the fabricated samples.
Language suggestions are listed in the comments above.
Author Response
Point 1: (line 76-79) More details of the epitaxial growth process are needed. What are the doping levels of the n-type and p-type GaN? What is the structure of the MQW?
Response 1: We thank the reviewer for the suggestion. In the revised manuscript, we have included information about the doping levels of n-type and p-type GaN, as well as the structure of the MQW.
As follows:
A 3.5μm-thick un-doped GaN (u-GaN) layer was grown first on the substrate. A 2μm n-GaN layer with a Si-doped concentration greater than 1019/cm3 was grown above the u-GaN layer and followed by a 10-period InGaN/GaN (3nm/12nm) multiple quantum wells (MQWs) with in composition of 13%. Finally, a 50nm-thick Mg-doped p-GaN with a doping concentration greater than 1019/cm3.
Point 2: (line 139) Please avoid using phrases like “it seems that…”
Response 2: Agreed. Modified in the revised manuscript.
As follows:
We can observe that the samples at position A always show a lower threshold of about 20% than those at position B.
Point 3: (line 138-147) I recommend the authors present these numeric values in the form of a table, listing the threshold, peak wavelength, FWHM, and the Q factor for each sample.
Response 3: Agreed. A parameters table has been included to the modified manuscript for easier understanding by common readers.
Point 4: (line 148) Typo? “later” -> “laser”
Response 4: We thank the reviewer for the suggestion. This means that in later simulations. We have made revise in the revised manuscript to make it easier to understand.
As follows:
All these lasing modes have been identified as WGM lasing in the later simulation results.
Point 5: (line 151) Could the authors provide a brief explanation and appropriate references for this statement? The readers may not be familiar with the effect of having tilted sidewalls.
Response 5: We thank the reviewer for the suggestion. Here, we add a simple explanation and provide appropriate references for this statement in the revised manuscript. For small microrod cavity structures, the tilt of the sidewall will make the light not well confined to the MQW plain, and reduce the overlap between the axial electric field and the active region, which will reduce the photon-electron coupling efficiency. At the same time, the light will randomly propagation to the bottom of the microrods, and the lasing wavelength will be more random [Ref. 16 in the revised manuscript ]. This makes it difficult to get a standard WGMs laser.
Point 6: (line 152) What does “regular and stable” mean? Please avoid using descriptive words. The statement needs to be scientifically precise and add values to the readers understanding of the subject matter.
Response 6: We thank the reviewer for the suggestion. The text is modified in revised manuscript according to reviewer suggestion.
As follows:
In figure 3, the lasing peaks do not shift under the varying excitation power density and always exist over the test range, which is consistent with the WGMs obtained by later FDTD simulations.
Point 7: (line 160) Please delete the word “obviously”
Response 7: Agreed. The word “obviously” is deleted in the revised manuscript.
Point 8: (line 170-224) The logic of this section needs clarification. There are 3 things involved in this discussion:
- Lasing modes of the MQW structures the authors fabricated.
- Lasing modes calculated by FDTD.
- TD distribution (Figure 5d).
We can only consider A as facts among these three, as it is experimentally measured. From my understanding of how the current draft is written, the authors made the assumption that C is true, then incorporated C in FDTD simulation and obtained results B. However, unless C is verified to be true, the comparison between A and B would be invalid. In other words, I understand that it makes sense that stronger lasing modes would correlate with regions with lower TD density, but we cannot say that TD density is “the” root cause. There could well be other factors that would affect lasing modes.
My suggestion: One way of making this discussion logically sound is by proving C a fact — For example, it can be proved by cross-sectional TEM imaging of the fabricated samples.
Response 8: We thank the reviewer for the suggestion. First of all, we agree with you. Due to experimental conditions, we are unable to augment TEM images for the time being. The use of PSS substrate technology to reduce the dislocation density generated during GaN epitaxial growth has been extensively studied, as has the growth process and dislocation behavior of GaN grown on PSS substrate, and the dislocation evolution process of GaN grown on PSS substrate shows consistency [Ref. 21-26 in the revised manuscript]. We also added references with TEM images [Ref. 21-24 in the revised manuscript] to illustrate the dislocation of GaN grown on PSS to demonstrate that the distribution of TDs was compatible with what we indicated, confirming C's validity. For example, as shown below. In the flat surface region the TDs extend directly to the GaN surface, on the tilted surface they bend and coalesced near the summit, finally extending along the top of the island to the surface.
TEM cross-section image of epitaxial GaN layers around a PSS [Ref. 23 in the revised manuscript].(Please see the attachment.)

Round 2
Reviewer 2 Report
good to proceed.
N/A.
Reviewer 3 Report
I appreciate the authors' careful and detailed response. Most of the comments are adequately addressed in the new manuscript.
On the last point of the previous comments (comment #8), I would be okay with the authors referencing previous works on this topic and adding a few sentences explaining how the dislocation distribution is formed, because GaN growth on PSS is well understood by the community, however, I would still like to remind the authors that this can be one of the common pitfalls that I see among researchers — Please be certain that one 1) completely understand and 2) have verified all the necessary assumptions that were made to construct the model, even though some may be well-studied in the past. Each experiment/sample is different. By treating every piece of past knowledge as known condition, one may be missing opportunities to new discovery, or stepping into the danger of building a theory upon false assumptions.